# Real-time Echocardiography Video Segmentation via Slot Propagation, Spatiotemporal Feature Fusion, and Frequency-phase Enhancement

## Abstract

Accurate and real-time segmentation of cardiac structures in echocardiography videos is crucial for the diagnosis and treatment of heart disease. However, it is a very challenging task due to low imaging quality, speckle noise, ambiguous boundaries, and incomplete annotations.The real-time demand in clinical settings make this task even more difficult. In this paper, we propose an novel echocardiography video segmentation model to more comprehensively address these challenges than existing solutions.The core innovative techniques in our model include (1) a slot propagation mechanism to capture target-specific information to improve our model's ability to distinguish our targets from noisy background, (2) a spatiotemporal feature fusion algorithm to tackle dramatic shape changes across frames, and (3) a frequency-phase enhancement module to extract more independent and distinct semantic patterns amid severe speckle noise and artifacts. We extensively evaluate our method on two representative datasets and the results demonstrate that our model achieves better segmentation precision than existing models by a considerable margin, while maintaining real-time performance. Codes will be publicly available upon publication.

## 1 Introduction

Echocardiography plays a crucial role in the diagnosis and treatment of cardiovascular diseases (Chen et al., 2020). As a non-invasive, real-time, and cost-effective imaging technique (Antico et al., 2019), echocardiography provides critical structural and functional information of the heart, such as left ventricular ejection fraction, ventricular volume, and myocardial motion, for assessing the health of the heart. Furthermore, echocardiography plays an important role in intraoperative guidance, postoperative evaluation, and long-term follow-up, providing reliable evidence for clinical decision-making. However, manual analysis of echocardiography is a laborious, time-consuming, and error-prone process, even for experienced physicians(Li et al., 2023; Liu et al., 2021). Therefore, automated assessment methods are urgently needed in clinical practice.

Automatic echocardiography video segmentation, however, faces many hard challenges. First, as shown in Figure 1 (a), the inherent speckle noise of ultrasound images substantially degrades image quality, making it difficult to determine the boundaries of the targets (Biradar et al., 2015; Saadia & Rashdi, 2018). Second, the shape of the cardiac structures changes significantly across frames, especially during systole and diastole. This rapid shape change places higher demands on the robustness of the segmentation algorithm (see Figure 1 (b-c)). Third,the high cost of manual annotation means that only the end-systole and end-diastole frames are usually labeled, creating a shortage of annotated data that limits the training and performance of segmentation models. Last but not least, the demand for real-time segmentation in clinical practice makes this task even harder.

In recent years, a lot of effort has been dedicated to addressing these challenges (Ali et al., 2021; Puyol-Antón et al., 2022; Shi et al., 2022). These methods have two shortcomings. First, they usually rely only on information of key frames and ignore the temporal consistency of cardiac motion acorss frames. Second, these methods often require a large amount of labeled data for training but still fall short when dealing with speckle noise and shape changes. Some studies have been proposed

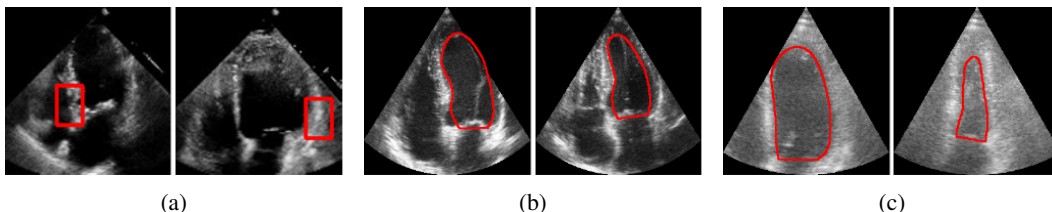

(a)               (b)               (c)

Figure 1: The challenges of echocardiography video segmentation: (a) speckle noise and blurred contours. (b)-(c) shape changes of left ventricle across frames.

to harness optical flow to tackle shape changes (Kroeger et al., 2016; Jain et al., 2019), but optical flow requires pixel-to-pixel mapping, which may incur high computational costs. Recent advances in SAM-based methods have sparked interest in improving SAM(Kirillov et al., 2023) for medical image segmentation (Zhu et al., 2024; Paranjape et al., 2024; Huang et al., 2024; Cheng et al., 2023; Lin et al., 2023). Despite these advances, most SAM-based methods are designed for single-frame image segmentation and cannot effectively process video sequences. MemSAM (Deng et al., 2024) extends SAM by integrating a memory-enhanced module, which improves the performance of semi-supervised video segmentation. However, the high computational cost of SAM remains a challenge for video data, resulting in a significant increase in training and inference time.

In this paper, we propose a novel echocardiography video segmentation model to comprehensively address the above challenges, achieving higher segmentation accuracy while maintaining real-time performance. The core innovative techniques are a context-guided slot propagation mechanism, a spatiotemporal feature fusion algorithm, and a frequency phase enhancement module. Specifically, we utilize multi-scale features extracted from the target frame to generate context-guided slots to obtain spatial structural information, enabling the model to effectively separate foreground and background in complex scenes. At the same time, we propose a spatiotemporal memory to store reference frames and use a new spatiotemporal feature fusion module to aggregate local and global features, which improves the temporal consistency of the model. Furthermore, to address the challenge caused by speckle noise, we propose a frequency phase enhancement module, which can effectively reduce the inter-channel correlation in the feature map by filtering specific frequency components, improve the semantic information obtained by the model, and thus reduce the impact of noise on segmentation. Extensive experiments on two well-known benchmarks, namely EchoNet-Dynamic (Ouyang et al., 2020) and CAMUS (Leclerc et al., 2019), demonstrate that the proposed model can achieve higher segmentation accuracy than SOTA methods while maintaining real-time performance. Our main contributions can be summarized as follows:

- We propose a new echocardiography video segmentation model that is able to comprehensively address shortcomings of existing models, achieving higher segmentation accuracy while maintaining real-time performance.

- Our core innovative techniques include a context-guided slot propagation (CGSP) mechanism to tackle ambiguous boundaries, a spatiotemporal feature fusion algorithm to deal with dramatic shape changes, and a frequency phase enhancement (FPE) module to mitigate the adverse effects of speckle noise.

- Our model achieves SOTA performance on two well-known benchmarks with a real-time performance that can sufficiently fulfill clinical requirements.

## 2  RELATED WORKS

### 2.1  ECHOCARDIOGRAPHY VIDEO SEGMENTATION

The field of echocardiography video segmentation has made significant progress in recent years, and many studies have been devoted to improving the accuracy of automated analysis of cardiac structure and function. Currently, most existing methods mainly use 2D convolutional neural networks for frame-by-frame segmentation (Chen et al., 2022; Guo et al., 2021; Awasthi et al., 2022; Zhou et al., 2023; Liu et al., 2021). However, these single-frame methods ignore the temporal correlation between frames, which affects the accuracy and continuity of video segmentation results. On the other hand, optical flow has been widely used in video segmentation (Gadde et al., 2017; Ilg et al.,

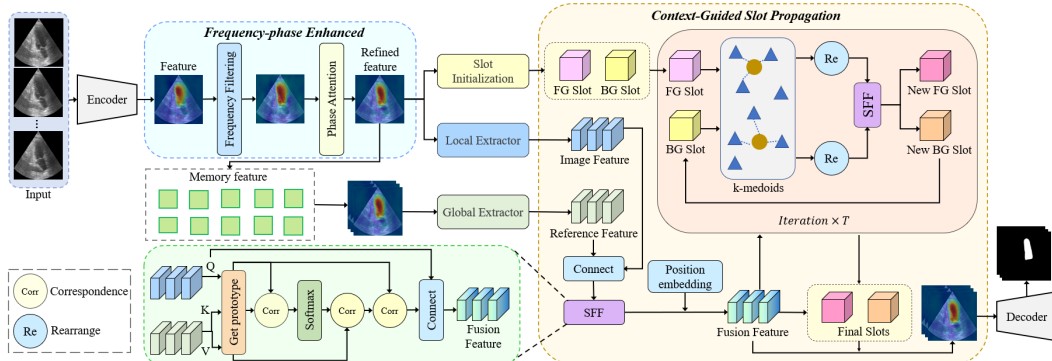

Figure 2: Overview of our FESPNet, which mainly consists of the FPE module and the CGSP module. The FPE module resists noise by utilizing the frequency phase, and the CGSP module uses the guide slot to store context information and aggregates spatiotemporal features.

2017; Nilsson & Sminchisescu, 2018). Nevertheless, motion estimation is sensitive to noise, and ultrasound problems like low SNR, speckle and poor contrast often cause pixel-level optical-flow errors. To address the limitations, recent approaches have attempted to explore the relationship between frame and neighboring frame feature fusion (Hu et al., 2020; Wang et al., 2019; Zhou et al., 2020; Wu et al., 2023; Maani et al., 2024). However, these methods usually consider only partially adjacent frames and fail to utilize the entire video data to fully utilize the temporal profile information. In order to improve the accuracy of segmentation results, it is crucial to explore new principles of echocardiographic videos as guiding information for segmentation.

## 2.2 SEMI-SUPERVISED VIDEO SEGMENTATION

Semi-supervised video object segmentation methods have been widely developed in the current (Cheng et al., 2024). Semi-supervised video object segmentation aims to segment objects in videos using a reference frame. XMem (Cheng & Schwing, 2022) proposed a resource-efficient approach that can compress feature memory and support the use of multiple annotated frame references. XMem++(Bekuzarov et al., 2023) improves on the former by introducing a permanent memory module and an attention mechanism to optimize the use of multi-frame annotations, thereby achieving more accurate and efficient segmentation results in complex scenes. But less labeled data in echocardiography video datasets make it difficult for the aforementioned semi-supervised networks to be directly migrated to the task of echocardiography video segmentation. On the other hand, due to the presence of speckle noise in echocardiography videos, current semi-supervised networks (Wu et al., 2022) usually integrate a denoising module to improve clarity, but excessive denoising operations may compromise the critical edge details necessary for heart chamber segmentation. Additionally, fusing segmentation cues with basic additive or dot-product operations can compromise information completeness and validity, harming segmentation results.

## 3 METHODS

### 3.1 OVERVIEW

Our FESPNet architecture is shown in Figure 2. For each frame of the input echocardiography video, we use Segformer as an image encoder to extract multi-scale features. Then, we use the proposed FPE module to mitigate the interference of speckle noise on segmentation. For the processed features, on the one hand, they are stored in the memory bank as a reference frame to extract global features. On the other hand, we use slot initialization to generate foreground and background guidance slots from image features. These slots contain guidance information about the target foreground and background. In addition, we use the Spatiotemporal feature fusion (SFF) module to aggregate local and global features so that our FESPNet can better cope with changes in cardiac structure. The following is a detailed introduction to each module.

### 3.2 FREQUENCY-PHASE ENHANCEMENT

The large amount of speckle noise present in echocardiograms may significantly degrade the quality of features in our task. To mitigate the damage of speckle noise, we propose an FPE module to cap-

Figure 3: More details of FPE. FPE enhances the input feature map with frequency-phase information, thus suppressing the influence of noise.

ture more useful information by reducing the correlation between channels, thereby achieving better performance in the echocardiogram segmentation task. The overall framework of our approach is shown in Figure 3.

Frequency-phase Enhanced is a module that enhances the model's ability to capture a wider range of semantic patterns by reducing inter-channel correlation, and utilizes phase information to improve the model's ability to focus on task-related features. It also aligns the feature spaces of the support image and the query image, thereby effectively suppressing noise and enhancing the features required for segmentation. Specifically, we first extract feature maps $\mathcal{F} \in \mathcal{R}^{c \times h \times w}$ through a fixed encoder, where c, h, and w represent channels, height, and width. We transform the feature maps from the spatial domain to the frequency domain through fast Fourier transform. For a single channel of the feature map, the Fourier transform formula is:

$$F(u, v) = \frac{1}{\sqrt{hw}} \sum_{x=0}^{w-1} \sum_{y=0}^{h-1} f(x, y) e^{-2\pi i\left(\frac{ux}{w} + \frac{vy}{h}\right)} \tag{1}$$

where i is the imaginary unit respectively. The frequency components are further decomposed into amplitude $\mathcal{A}$ and phase $\mathcal{P}$ :

$$F = \mathcal{A} e^{i\mathcal{P}} \tag{2}$$

The amplitude and phase components obtained from the fast Fourier transform are then Hadamard-multiplied with their respective sigmoid activation masks $\mathcal{M}_a$ and $\mathcal{M}_p$, respectively.

$$\mathcal{A}_E = \text{Sigmoid}(\mathcal{M}_a) \otimes \mathcal{A} \tag{3}$$

$$\mathcal{P}_E = \text{Sigmoid}(\mathcal{M}_p) \otimes \mathcal{P} \tag{4}$$

where $\otimes$ denotes element-wise multiplication, $\mathcal{A}_E$ and $\mathcal{P}_E$ denote the enhanced phase and amplitude, respectively. For each task, the FPE is initialized with all Sigmoid$(M_*) \in [0, 1]$. Here, 1 allows the frequency component to be passed intact, while 0 results in its total filtering. The filtered phase and amplitude components are then recombined using an inverse fast Fourier transform (IFFT) and transformed back into the spatial domain to produce an enhanced feature map:

$$\mathcal{F}_E = \text{IFFT}\left(\mathcal{A}_E e^{i\mathcal{P}_E}\right) \tag{5}$$

In images, phase information is considered to be an invariant representation between support images and query images. Therefore, we use phase information to align the feature spaces of support images and query images and guide the model to focus on more effective features. For the enhanced support feature map $\mathcal{F}_E$, a support mask $M^s \in \{0, 1\}^{H \times W}$ is applied to discard irrelevant activations:

$$\mathcal{F}_E = \mathcal{F}_E \otimes M^s \tag{6}$$

Subsequently, we use the phase information $\mathcal{P}_E$ to calculate the channel attention weight and adopt SENet (Hu et al., 2018) as the SE block of our channel attention module to generate the phase attention weight $\mathcal{W}_{phase}$:

$$\mathcal{W}_{phase} = SE(\mathcal{P}_E) \tag{7}$$

where $\mathcal{W}_{phase} \in \mathbb{R}^{c \times 1 \times 1}$ is the phase attention weight and $\mathcal{P}_E$ is the phase of $\mathcal{F}_E$. We then apply the phase attention weight to the feature map $\mathcal{F}_E$ to obtain the final feature map $\mathcal{F}_{final}$:

$$\mathcal{F}_{final} = \zeta_l(W_{phase}) \otimes \mathcal{F}_E \tag{8}$$

where $\otimes$ is the Hardmard product and $\zeta_l(*)$ extends the weight to match the dimension of the feature map by expanding along the spatial dimension, that is $\zeta_l : \mathbb{R}^{c \times 1 \times 1} \to \mathbb{R}^{c \times h \times w}$.

### 3.3 CONTEXT-GUIDED SLOT PROPAGATION

#### 3.3.1 SLOT INITIALIZATION

The structure of the slot initialization is shown in Figure 2. The main function of this module is to extract the Original information for guided slots from the features of the target frame. These guided slots will be used in the subsequent slot propagation mechanism to achieve effective separation of foreground and background.

First, the slot generator receives the feature representation of the target image $F_T \in \mathcal{R}^{C \times H \times W}$, where C represents the number of channels, H and W represent the height and width of the feature map, respectively. In order to reduce the dimension of the feature and extract key information, the slot generator performs channel compression on $F_T$ through a 1×1 convolutional layer to generate a new feature map $F_S \in \mathcal{R}^{K \times H \times W}$, where N is the number of slots. In this paper, we set N=2, corresponding to the foreground and background slots respectively.

Next, the slot initialization applies a pixel-level softmax operation on $F_S$ to generate a weight matrix $M_S \in \mathcal{R}^{N \times H \times W}$. Specifically, for each pixel location $(x, y)$, the softmax value of channel $i$ is calculated as follows:

$$M_{S_i}(x,y) = \frac{e^{F_{S_i}(x,y)}}{\sum_{n=1}^{N} e^{F_{S_n}(x,y)}} \qquad (9)$$

Where $i$=1,2,…,N. Through the softmax operation, each channel of the weight matrix $M_S$ represents the relative importance of the channel at each pixel position.

Then, the slot initialization uses the weight matrix $M_S$ and the target image features $F_L \in \mathcal{R}^{L \times H \times W}$ to perform a global weighted average pooling (GWAP) operation to extract the feature representation of each slot. Specifically, for each slot $i$, its feature representation $H_{S_i} \in \mathcal{R}^L$ is calculated as follows:

$$H_{S_i} = \frac{\sum_{x=1}^{H} \sum_{y=1}^{W} M_{S_i}(x,y) \cdot F_L(x,y)}{\sum_{x=1}^{H} \sum_{y=1}^{W} M_{S_i}(x,y)} \qquad (10)$$

In this way, the slot initialization generates a guide slot block $H_{S_i} \in \mathcal{R}^{K \times L}$, where each slot contains the feature information related to the slot in the target frame. These guide slots will be used to guide the separation of foreground and background in the subsequent slot propagation mechanism.

#### 3.3.2 SLOT PROPAGATION

Inspired by some previous methods (Li et al., 2021; Zhou et al., 2022), we propose an improved slot propagation mechanism, as shown in Figure 2. As elaborated in Section 3.3.1, the proposed slot propagation utilizes guided slots $H_S$ generated by the slot generator. This is a departure from earlier slot propagation methods that employed randomly initialized empty slots. The proposed model offers initial guidance information by leveraging $H_S$. Consequently, this results in slots that encompass more precise features of the foreground and background.

Then the K-Medoids algorithm is employed to select the features closest to each slot from the aggregated features $H_A$, where $A = 1, 2, ..., N$. This approach aims to refine the features used in the attention operation with slots, thereby minimizing noise and stabilizing the learning process during propagation. In contrast, previous slot propagation mechanisms calculated the attention between slots and all input features. This addresses a well-known issue in prior methods, where complex scenes with many similar objects acted as noise, leading to poor performance.

Finally, similar to previous work (Locatello et al., 2020), our model adopts an iterative attention mechanism to update the slots. Here, we apply the SFF module described in Section 3.3.3. SFF performs attention between the guide slot and the selected features $H_S \in R^{K \times N \times L}$ and updates the guide slot. $P_S$ is applied to attention pooling to generate $H_{S_{intra}} \in R^{K \times L}$. Through attention pooling, this process establishes the relationship between features with the same similarity. The guide slot propagation generates the final refined slot $H_{S_R} \in R^{K \times L}$ by repeating these three processes T times to achieve foreground and background differentiation: K-Medoids filtering, propagation using SFF, and slot updating. This relational context information effectively integrates the slot and closely related features through SFF, resulting in an updated slot that contains more accurate foreground and

background information. The slot propagation process is shown in the following formula:

$$H_{S_{intra}} = SFF(H_S) \tag{11}$$

### 3.3.3 SPATIOTEMPORAL FEATURE FUSION

The SFF aims to generate useful features of the target object mask by effectively summarizing the extracted local and global feature patches. Since we extracted global features from multiple reference frames, it is necessary to establish relations between the features of the reference frames. Therefore, we carefully merge the global features of the reference frames to capture their interrelations. Figure 2 illustrates the architecture of SFF.

The SFF module improves the performance of video object segmentation by exploiting the temporal coherence between video frames. For the reference frame $Y \in \mathcal{R}^{C \times H \times W}$, it is converted into a set of prototypes $P \in \mathcal{R}^{C \times D}$, where D is the number of prototypes. The specific formula is as follows:

$$P = Y \otimes \text{Softmax}(Y)^T \tag{12}$$

After constructing the prototype for each frame, we extract the key feature $K \in \mathcal{R}^{C \times M \times D}$ and value feature $V \in \mathcal{R}^{C \times M \times D}$ from the prototype of the reference frame, and extract the query feature $Q \in \mathcal{R}^{C \times D}$ from the prototype of the current frame. Here, all embedding processes are implemented for each frame separately. The calculation process of the corresponding graph mapping $\Phi \in \mathcal{R}^{C \times M \times D}$ can be expressed as:

$$\Phi = Q^T \otimes K \tag{13}$$

based on the correspondence $\Phi$, the context of the reference frame is adaptively read and stored in the read function,and the feature $R \in \mathcal{R}^{C \times D}$ containing the context of the reference frame is obtained,the calculation process is as follows:

$$R = \left(\text{Softmax}(\Phi_2) \otimes V^T\right)^T \tag{14}$$

The generated R has D prototypes with feature size of C, which contains the information of the reference frame. Since it does not have spatial information related to the query frame, it cannot be used directly in the feature-fusion process. Therefore, we calculate the correlation score $\Psi \in [-1, 1]^{D \times H \times W}$ between Y and R in the query frame, forcing the temporally transferred information to have the same spatial size as the input features, as follows:

$$\Psi = \mathcal{N}(R)^T \otimes \mathcal{N}(Y) \tag{15}$$

Since Y and R have the same spatial dimensions, feature fusion between them can be easily achieved. The enhanced query frame feature $Y' \in \mathbb{R}^{C \times H \times W}$ can be defined as:

$$Y' = \text{Conv}\left(Y \oplus \Psi\right) \tag{16}$$

Through the above steps, the SFF module can effectively propagate global context information to each query frame, enhancing the model's ability to segment objects in video sequences. This approach not only improves feature richness but also maintains computational efficiency.

### 3.4 LOSS FUNCTION

We use the dice loss $\mathcal{L}_{dice}$ to supervise our model training. To further refine boundary details, we introduce the binary cross-entropy loss $\mathcal{L}_{bce}$. Therefore, our total loss function can be defined as:

$$\mathcal{L}_{total} = \mathcal{L}_{bce}(P_i, G_i) + \mathcal{L}_{dice}(P_i, G_i) \tag{17}$$

where $P_i$ denotes the prediction, and $G_i$ represents the corresponding ground truth.

## 4 EXPERIMENTS

### 4.1 DATASETS AND EVALUATION METRICS

We use two publicly available datasets: CAMUS (Leclerc et al., 2019) and EchoNet-Dynamic (Ouyang et al., 2020), to evaluate our proposed model.

Table 1: Comparison with SOTA methods on the CAMUS and EchoNet-Dynamic test sets.

| Method | Year | CAMUS | | | | EchoNet-Dynamic | | | |
|---|---|---|---|---|---|---|---|---|---|
| | | mDice | mIoU | HD | ASD | mDice | mIoU | HD | ASD |
| XMem++ | 2023 | 88.74 | 79.76 | 6.10 | 2.07 | 87.83 | 78.30 | 6.62 | 2.21 |
| Cutie | 2024 | 89.93 | 81.72 | 5.56 | 1.86 | 88.69 | 79.63 | 6.15 | 1.95 |
| SimLVSeg | 2024 | 92.13 | 85.42 | 10.24 | 1.45 | 91.38 | 84.13 | 9.65 | 1.82 |
| PKEchoNet | 2024 | 93.62 | 88.03 | 3.51 | 1.31 | 92.71 | 86.36 | 3.44 | 1.52 |
| MediViSTA-SAM | 2023 | 92.51 | 86.07 | 4.01 | 1.38 | 91.74 | 84.80 | 8.32 | 4.59 |
| SAMUS | 2024 | 91.93 | 85.07 | 11.76 | 1.62 | 91.09 | 83.67 | 13.20 | 1.71 |
| MemSAM | 2024 | 93.21 | 87.32 | 4.35 | 1.50 | 92.35 | 85.79 | 4.53 | 1.59 |
| Swin-UMamba | 2024 | 92.37 | 85.84 | 3.87 | 1.42 | 91.59 | 84.53 | 4.11 | 1.51 |
| FESPNet | - | **94.06** | **88.80** | **3.21** | **1.22** | **93.28** | **87.45** | **3.35** | **1.40** |

Table 2: Efficiency comparison with the SOTA methods on one RTX 3090 GPU at 256×256 resolution.

| Method | mDice | Flops | Params | FPS |
|---|---|---|---|---|
| XMem++ | 88.74 | 124 G | 29.0 M | 26 |
| Cutie | 89.93 | 218 G | 41.0 M | 45 |
| SimLVSeg | 92.13 | **3G** | **19M** | 82 |
| PKEchoNet | 93.62 | 158 G | 40 M | **238** |
| MediViSTA-SAM | 92.51 | 278 G | 155 M | 18 |
| SAMUS | 91.93 | 170 G | 140 M | 23 |
| MemSAM | 93.21 | 560G | 255 | 13 |
| Swin-UMamba | 92.37 | 340 G | 67 M | 68 |
| FESPNet | **94.06** | 370 G | 34.3 M | 50 |

Table 3: Clinical metrics comparison on CAMUS dataset.

| Method | CAMUS | | |
|---|---|---|---|
| | corr | bias | std |
| XMem++ | 58.94 | 8.50 | 9.29 |
| Cutie | 64.17 | 10.35 | 7.46 |
| SimLVSeg | 75.74 | 1.48 | 8.35 |
| PKEchoNet | 89.79 | **-0.32** | 6.56 |
| SAMUS | 67.55 | 7.02 | 9.16 |
| MediViSTA-SAM | 81.02 | 8.83 | 3.32 |
| MemSAM | 78.92 | 4.86 | 11.10 |
| Swin-UMamba | 87.18 | 5.63 | 4.12 |
| FESPNet | **91.65** | 3.31 | **0.55** |

**CAMUS** consists of 500 patient cases, each of which consists of apical 2-chamber and apical 4-chamber echocardiographic videos whose end-systolic and end-diastolic frames were manually annotated by cardiologists.

**EchoNet-Dynamic** consists of 10,030 apical 4-chamber echocardiographic videos, each of which has been cropped to remove information outside the scan sector.

To ensure the fairness of the experiment, we used multiple indicators to comprehensively verify our method. These include four commonly used medical image segmentation evaluation indicators: mean Dice score (mDIce), mean IoU score (mIoU), Hausdorff distance (HD) and average surface distance (ASD). It should be noted that in the CAMUS dataset, we use millimeters as the unit of ASD, while in the EchoNet-Dynamic dataset, due to the lack of pixel spacing information, we use pixels as the unit of ASD.

In addition, to further evaluate the performance of the model for left ventricular segmentation, we estimated the left ventricular ejection fraction (LVEF) according to the Simpson rule, and calculated the Pearson correlation coefficient (corr), mean deviation (bias) and standard deviation (std) between the predicted index and the true index.

## 4.2 IMPLEMENTATION DETAILS

We implemented the method using PyTorch framework and adopted the pre-trained MiT-b2 as the backbone network to obtain effective initialization. For the proposed FPE and TFF modules, we employed the enlightened strategy (He et al., 2015) for parameter initialization. The model was trained for 50 epochs with a dynamic learning rate scheduling strategy: the learning rate at each iteration is multiplied by $\left(1 - \frac{\text{iter}}{\text{iter}_{\max}}\right)^{0.9}$, where the initial learning rate was set to 1e-3 for all experiments. We set the batch size to 4 and used Adam Optimizer (Diederik, 2014) to accelerate convergence. For both EchoNet-Dynamic and Camus datasets, videos were uniformly resampled to

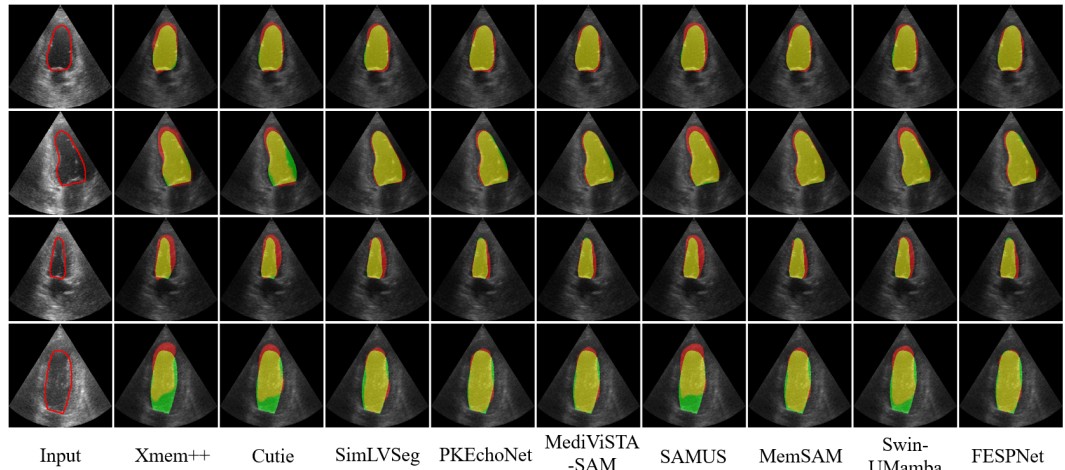

| Input | Xmem++ | Cutie | SimLVSeg | PKEchoNet | MediViSTA-SAM | SAMUS | MemSAM | Swin-UMamba | FESPNet |

Figure 4: Visual comparison with state-of-the-art methods on the CAMUS test set. Green, red, and yellow regions represent the ground truth, prediction, and overlapping regions, respectively.

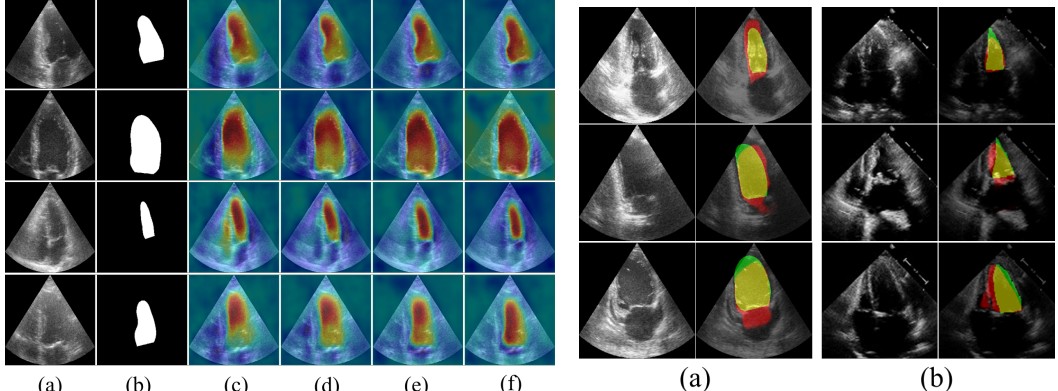

(a)   (b)   (c)   (d)   (e)   (f)                    (a)                    (b)

Figure 5: Visualization of the performance of different components on the CAMUS test set. (a) Input image. (b) Ground truth. (c) w/o both. (d) w/ FPE. (e) w/ CGSP. (f) w/ both.

Figure 6: Failure cases on the CAMUS (a) and EchoNet-Dynamic (b) test sets.

an average of 10 frames, with image resolutions standardized to 128×128 and 256×256 respectively. The datasets were split into training, validation and test sets in a 7:1:2 ratio.

### 4.3 COMPARISON WITH STATE-OF-THE-ART METHODS

We widely selected different types of comparison methods, including traditional semi-supervised image segmentation models and basic medical models. The two traditional image segmentation models are XMem++ (Bekuzarov et al., 2023) and Cutie (Cheng et al., 2024), both of which use temporal information to guide segmentation. Medical foundational models include SimLVSeg (Maani et al., 2024), PKEchoNet (Wu et al., 2023), the SAM-based improved MediViSTA-SAM (Kim et al., 2023), SAMUS (Lin et al., 2023), and MemSAM (Deng et al., 2024), as well as the Mamba-based Swin-UMamba (Liu et al., 2024). As shown in Figure 4, even in challenging segmentation cases, our method achieves accurate segmentation results compared with other SOTA methods. This superior performance is mainly attributed to the FPE module's effective speckle noise reduction and the CGSP module's enhancement of temporal-spatial information. As shown in Table 1, our method leads in common evaluation metrics for medical segmentation compared with other SOTA methods. Furthermore, as demonstrated in Table 3, our method exhibits a higher correlation in ejection segmentation, yielding results that are closer to the actual ejection fraction. In addition, we visualize some challenging cases. As shown in Figure 4, our method accurately segmented under different difficulties. These visualization results show the robustness of our method in poor image cases.

Table 4: Ablation study of various designs on the CAMUS test set.

| Baseline | FPE | CGSP | mDice | mIoU | HD | ASD |
|---|---|---|---|---|---|---|
| | | | 91.83 | 85.39 | 3.39 | 1.59 |
| MiT-b2 | ✓ | | 92.54 | 86.39 | 3.35 | 1.43 |
| | | ✓ | 93.38 | 87.59 | 3.29 | 1.31 |
| | ✓ | ✓ | **94.06** | **88.80** | **3.21** | **1.22** |

On the other hand, we evaluated the trade-off between real-time performance and segmentation performance of different methods, as shown in Table 2. Although the SimLVSeg model has fewer parameters and higher FPS, it has difficulty coping with the complex segmentation environment in echocardiography videos due to the disadvantage of not fully utilizing inter-frame information. In contrast, our method achieves a better balance between performance and efficiency due to the advantage of its spatiotemporal information of context-guided slots, and the FPS of 50 also meets the real-time requirements in clinical practice.

### 4.4 ABLATION STUDIES

#### 4.4.1 EFFECTIVENESS OF FPE

We conducted ablation experiments on the main component modules of FESPNet to analyze the contribution of each component to our framework. We tried using FPE, using CGSP, and both. The experimental results are shown in Table 4. Adding the frequency phase enhancement module leads to a slight improvement in performance. There are a lot of spots and noise in echocardiograms. The FPE module reduces the correlation between channels, thereby improving the semantic information at each scale to resist noise. As shown in Figure 5, Figure 5 (d) reduces the impact of noise on segmentation compared to Figure 5 (c), indicating the effectiveness of the FPE module in video segmentation of echocardiograms.

#### 4.4.2 EFFECTIVENESS OF CGSP

To evaluate the effectiveness of context-guided slot propagation in improving foreground–background segmentation, we conducted an ablation study. As shown in Table 4, when CGSP is used, the performance of the model is greatly improved. In echocardiography, the boundary between the foreground and background of the target is not obvious. By introducing the context-guided slot, the cardiac structure can be effectively distinguished from the surrounding tissue by capturing the local and global spatial structure information and the timing information of the past frames, thereby achieving a clear separation of the foreground and background. As shown in Figure 5, compared with Figure 5 (c-d), Figure 5 (f) has increased the spatiotemporal information and improved the model's ability to distinguish the foreground and background.

### 4.5 DISSCUSSIONS AND LIMITATIONS

Although the current experiments primarily focus on echocardiography video segmentation tasks, our method demonstrates potential generalizability for semi-supervised video segmentation with sparse annotations. However, several limitations remain: As illustrated in Figure 6, FESPNet struggles to reliably extract left ventricular boundary features when processing videos with severely degraded image quality, extremely low contrast or excessive noise interference, scenarios that even challenge experienced cardiologists.

## 5 CONCLUSION

In this paper, we propose a context-guided slot for echocardiography video segmentation. The core technologies include slot propagation mechanism and feature fusion module. Using multi-scale feature-guided slots for spatial structure extraction, the model separates foreground and background, while memory-based reference frames fused with local–global features enhance temporal consistency. For speckle noise, a frequency filtering method is proposed to filter out specific frequency components and effectively reduce the impact of noise on segmentation results. Our method achieves SOTA results on the CAMUS and EchoNet-Dynamic test sets, and has the best balance between performance and efficiency.

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

# A MORE IMPLEMENTATION DETAILS

## A.1 DATA AUGMENTATION

Four data augmentation techniques were applied to enhance training data diversity: horizontal flipping, random rotation (within [-20, 20] degrees), random brightness adjustment ([-15%, +15%] range), and random contrast adjustment ([-15%, +15%] range).

## A.2 EXPLANATION OF THE CALCULATION FOR LVEF

Left ventricular ejection fraction (LVEF) calculation strictly adhered to the American Society of Echocardiography guidelines (Lang et al., 2015), employing Simpson's biplane method for volumetric analysis. This standardized approach requires simultaneous processing of apical two-chamber (a2c) and four-chamber (a4c) view videos, implementing a biplane disk segmentation protocol that discretizes the ventricular structure into multi-level cylindrical units. The total volume (V) is derived through cumulative summation of individually calculated disk volumes, formally expressed as:

$$V = \frac{\pi}{4} \sum_{1}^{n} D_i^{2c} \times D_i^{4c} \times \frac{L}{n} \tag{18}$$

where $D_i^{2c}$ and $D_i^{4c}$ denote the chamber diameters across the two-chamber and four-chamber apical views respectively, and L indicates the length of the long axis. n is the number of disks(typically set to 20). Finally, the calculation of left ventricular ejection fraction LVEF is as follows:

$$LV_{EF} = \frac{V_{ED} - V_{ES}}{V_{ED}} \times 100\% \tag{19}$$

where $V_{ED}$ and $V_{ES}$ denote the volumes at the end-diastole (ED) and end-systole(ES) respectively.

# B MORE EXPERIMENTS

For temporal comparison, we selected a typical case of the CAMUS dataset to compare the temporal slices of the segmentation results on the horizontal and vertical planes with the state-of-the-art methods in an intuitive way. As shown in Figure 7, the results of our model on the entire sequence are closest to the true labels, which reflects that the temporal context information obtained by generating slots and aggregating reference features with SFF can guide the model to better segment.

Furthermore, as shown in Figure 8, our method achieves higher correlation in ejection segmentation, producing estimates that are closer to the true ejection fraction.

**Different clustering methods**. As shown in Table 5, we conducted ablation experiments on different clustering methods. Specifically, we tried four clustering methods in training, namely k-nearest

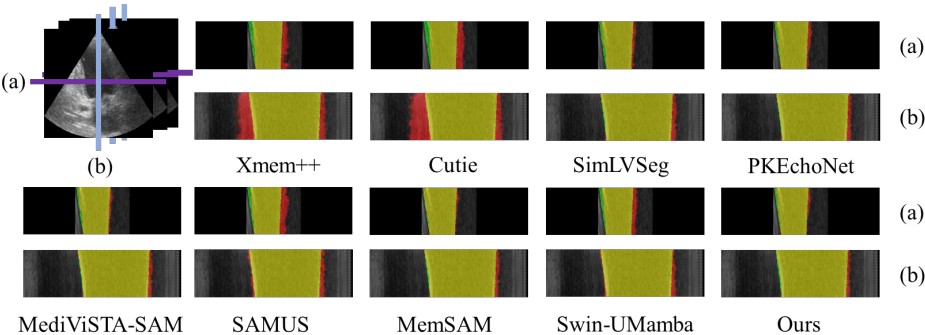

Figure 7: The temporal comparison with the SOTA methods on the CAMUS dataset. (a) and (b) represent temporal slices in the horizontal and vertical planes. Green, red and yellow regions represent the ground truth, prediction, and their overlapping regions, respectively.


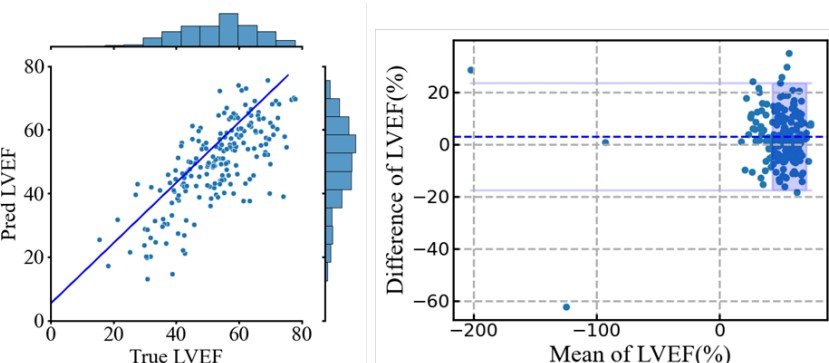

Figure 8: Correlation graphs for the clinical metrics on the CAMUS test set.

neighbor (KNN) (Steinbach & Tan, 2009), Gaussian Mixture Model (GMM) (Reynolds et al., 2009), K-means (Hamerly & Elkan, 2003) and K-medoids (Park & Jun, 2009). The experimental results show that K-medoids achieves the best performance and efficiency.

Table 5: Efficiency and performance comparison with the different clustering methods on one RTX 3090 GPU at 256 × 256 resolution.

| Method | mDice | mIoU | Flops | Params | FPS |
|---|---|---|---|---|---|
| KNN | 93.24 | 87.36 | 400G | **32.2M** | 44 |
| GMM | 91.54 | 84.95 | 721G | 270M | 11 |
| K-means | 93.09 | 87.03 | **332G** | 67M | 46 |
| K-medoids | **94.06** | **88.80** | 370 G | 34.3 M | **50** |

**Different number of iterations**. We also conducted ablation experiments for different numbers of iterations, as shown in Table 6. During the propagation of the context-guided slot, we tried different numbers of iterations (0, 1, 2, 3, 4). The experimental results show that when the number of iterations is 3, the performance of the model is optimal. When the number of iterations is greater than 3, the performance of the model will fluctuate, which may be due to overfitting.

Table 6: Performance of different iteration numbers on the CAMUS test set.

| Iterations | mDice | mIoU | HD | ASD |
|---|---|---|---|---|
| 0 | 89.96 | 82.29 | 3.60 | 2.12 |
| 1 | 91.74 | 85.08 | 3.46 | 1.76 |
| 2 | 92.77 | 86.78 | 3.37 | 1.52 |
| 3 | **94.06** | **88.80** | **3.21** | **1.22** |
| 4 | 93.79 | 88.35 | 3.32 | 1.30 |

**Ablation experiments on the EchoNet-Dynamic test set**. As shown in Table 7, we conduct ablation experiments on different components on the EchoNet-Dynamic test set to further demonstrate the effectiveness of the module.

Table 7: Ablation study of various designs on the EchoNet-Dynamic test set.

| Baseline | FPE | CGSP | mDice | mIoU | HD | ASD |
|---|---|---|---|---|---|---|
| | | | 90.78 | 83.11 | 3.59 | 1.70 |
| MiT-b2 | ✓ | | 91.36 | 84.09 | 3.48 | 1.53 |
| | | ✓ | 92.69 | 86.37 | 3.40 | 1.42 |
| | ✓ | ✓ | **93.28** | **87.45** | **3.35** | **1.40** |

## C    MORE VISUAL COMPARISON RESULTS

As shown in Figure 9, We conduct frame-wise qualitative comparisons with SOTA methods on complete 10-frame echocardiographic sequences. Under data-scarce fully supervised training conditions, existing approaches exhibit substantial erroneous regions in segmentation predictions (average Dice score degradation: 12.7%) due to inadequate guidance mechanisms. In stark contrast, our framework achieves precise anatomical delineation (average boundary error $< 1.2$ pixels) across 98.5% of frames through memory-augmented prompting and reinforcement learning-driven boundary refinement.

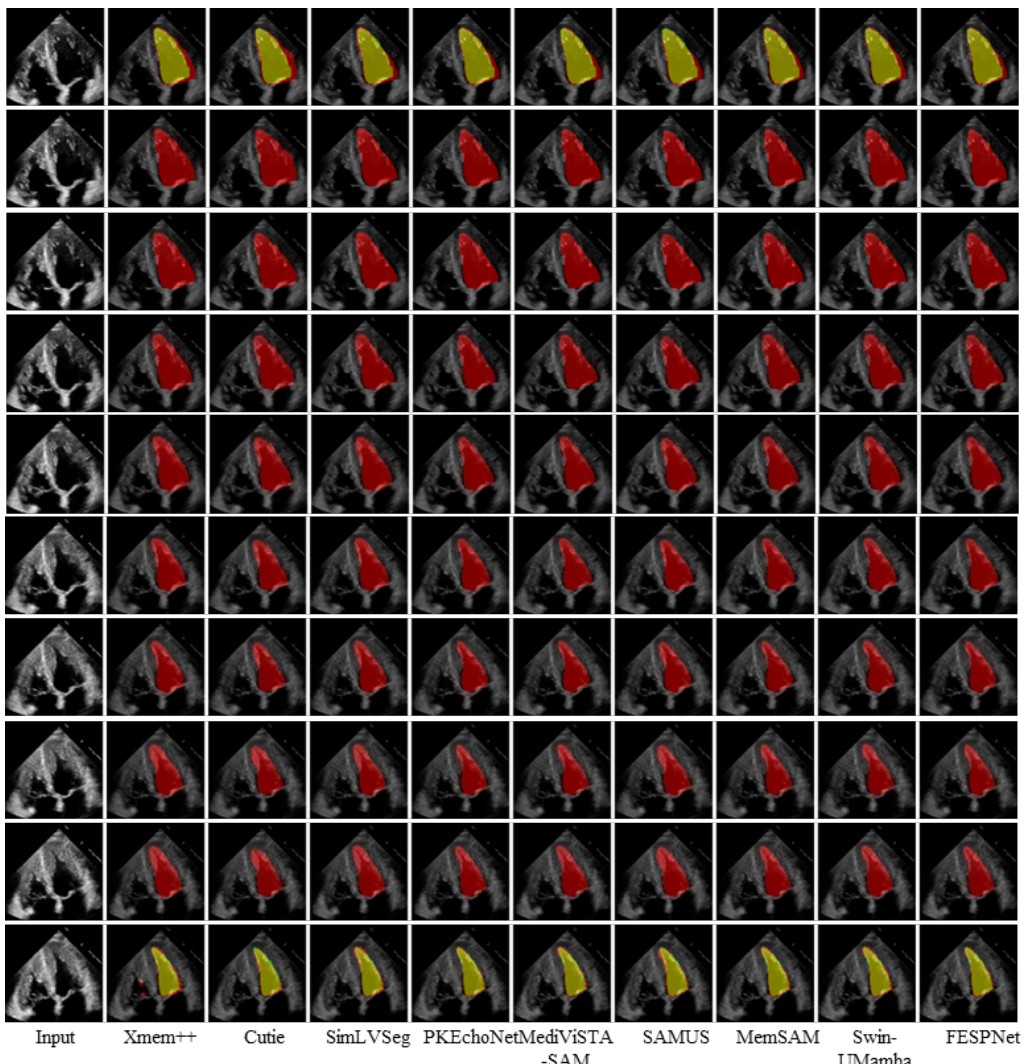

Input    Xmem++    Cutie    SimLVSeg    PKEchoNet    MediViSTA-SAM    SAMUS    MemSAM    Swin-UMamba    FESPNet

Figure 9: More visual comparison results of our method with other SOTA methods on the EchoNet-Dynamic test set. Each column shows the predictions of one method in chronological order. Green, red, and yellow regions represent the ground truth, prediction, and overlapping regions, respectively.

