# OpenReview forum: "Real-time Echocardiography Video Segmentation via Slot Propagation, Spatiotemporal Feature Fusion, and Frequency-phase Enhancement"
_ICLR.cc/2026/Conference — Submitted to ICLR 2026_

### Official Review · Reviewer_7Pap · 2025-10-31

**Soundness:** 2
**Presentation:** 2
**Contribution:** 2
**Rating:** 2
**Confidence:** 4

**Summary:**

This paper presents a method for the segmentation of echocardiography video. The method introduces three main components, including slot propagation to distinguish targets from noisy background, a spatiotemporal feature fusion to capture temporal information, and a frequency-phase enhancement module to extract semantic patterns. Experiments are conducted on two public echocardiography video datasets.

**Strengths:**

The paper is clearly written and easy to follow.

**Weaknesses:**

- The methodology lacks sufficient novelty. The proposed approach is a straightforward combination of three components, each of which has already been explored in prior video segmentation studies. The paper does not clearly demonstrate how this integration leads to new insights.

- Table 1 shows that the method obtains only marginal performance gains over existing approaches, with merely 0.44% and 0.57% improvement in Dice on the two datasets. These minimal improvements further suggest that the contribution may primarily come from an empirical combination of existing techniques rather than a fundamentally new idea.

- The paper emphasizes real-time capability as an important aspect of echocardiography segmentation. However, this claim is not well supported by the experimental evidence. In Table 2, the FPS of the proposed method is significantly lower than the second-best method PKEchoNet, and the Flops of the proposed method are even higher than some SAM-based methods, undermining the claim of computational efficiency.

- The experimental evaluation is limited to only two echocardiography datasets. This restricted scope raises concerns about the generalizability of the proposed approach and limits its potential interest for a broader audience.

**Questions:**

- Could the authors clearly describe the specific novelty or unique contribution of their approach beyond the straightforward integration of the three existing components?

- Can the authors demonstrate practical benefits or downstream impact that justify the contribution despite the small numerical differences?

- Could the authors clearly justify the computational advantage of the method?

- Could the method generalize to other tasks or datasets? What adaptations would be required?

---

### Official Review · Reviewer_Z77F · 2025-10-31

**Soundness:** 2
**Presentation:** 2
**Contribution:** 3
**Rating:** 2
**Confidence:** 5

**Summary:**

This work introduces a method to segment LV in echocardiography. The proposed model contains a Frequency-Phase Enhancement and Context-Guided Slot Propagation modules, which handle noise and temporal consistency in echocardiography videos. They demonstrated reasonable performance on two well-known datasets.

**Strengths:**

They introduced this combination of the Frequency-Phase Enhancement module and the Context-Guided Slot Propagation, which seems a good way to handle noise and maintain temporal consistency in echocardiography videos. Another possible strength is that they do show strong quantitative results on the datasets they used, like outperforming several existing methods on those benchmarks. The authors presented several interesting ablation studies.

**Weaknesses:**

1-	There is a discrepancy in the reported Dice scores for other methods. For instance,  SimLVSeg reported an average dice of 93.32 on the EchoNet dataset, where the value is reported as 91.38 in Table 1. That’s a major problem if they’re citing different numbers from the original paper without explaining why.
2-	The introduced work is over-engineered and complex and hence may raise the issue of generalizability. They only evaluated on the CAMUS and EchoNet-Dynamic datasets, so the results might not extend to other datasets like HMC-QU or ACDC. That’s a limitation on how widely applicable their findings are.
3-	Some ablations are done on the much smaller dataset CAMUS e.g., the number of iterations. This isn’t ideal as they could have used the larger dataset, EchnoNet, for those tests or both datasets to give a more balanced view.
4-	No standard deviations are reported in their performance metrics, which makes it hard to know how consistent or variable their results are. And it would have been better to highlight the larger dataset results in the main text instead of the smaller dataset. I would switch tables 4 and 7.
5-	The complexity of the model is not clear. The real-time performance claim might depend heavily on specific hardware, and the authors didn’t elaborate on that. So it might not be as generalizable in terms of real-time use on different systems.

**Questions:**

Please address the 5 points in weaknesses section.

---

### Official Review · Reviewer_vDtk · 2025-11-01

**Soundness:** 2
**Presentation:** 3
**Contribution:** 2
**Rating:** 4
**Confidence:** 4

**Summary:**

This paper presents a technically sound echocardiography segmentation model that addresses three critical challenges in cardiac ultrasound analysis through well-motivated innovations: the Context-Guided Slot Propagation (CGSP) mechanism for ambiguous boundary handling, spatiotemporal feature fusion for managing cardiac shape variations, and Frequency Phase Enhancement (FPE) for speckle noise reduction.

**Strengths:**

- paper well written
- clinical motivation significant
- great segmentation performance

**Weaknesses:**

-  Insufficient Ablation on FPE Components. Difficult to assess the FPE module's design effectiveness and understand which frequency components (amplitude or phase) are more critical for noise suppression in echocardiography. e.g. using only phase to add weights, okay but why?
- Same to FPE, CGSP lacks of ablation and techinical motivation for specific design.
- PKEchoNet seems to have exellent performance as well as FPS, while this method has relatively small gain but large FPS drop. The significance of the paper needs to be further justified.

**Questions:**

- in FPE, for Ma and Mp, the paper only states “Sigmoid(M∗) ∈[0, 1]” which is trivially true. But how are Ma and Mp be initialized? (e.g., uniform across all frequencies vs. noise-informed initialization) Does it influence the finial results?
- Again, how was Ms initialized specifically?
- Are both phase and amplitude filtering equally important? An ablation comparing (a) amplitude-only filtering, (b) phase-only filtering, and (c) joint filtering would clarify the individual contribution of each component and whether separate masks are necessary.
- How does FPE compare against simpler spatial-domain denoising methods (e.g., learnable convolutions, attention mechanisms, or non-local means)? Is the frequency-domain approach demonstrably superior, or does it add unnecessary complexity?
- In slot initialization, is N=2 sufficient? The paper mentions "foreground and background slots" suggesting N=2, but cardiac structures have multiple regions.
- Why use K-Medoids specifically for feature selection?
- How is T (number of propagation iterations) determined?

---

### Official Review · Reviewer_72j4 · 2025-11-01

**Soundness:** 2
**Presentation:** 3
**Contribution:** 3
**Rating:** 2
**Confidence:** 4

**Summary:**

While the paper addresses an important and clinically relevant problem, the proposed method however lacks sufficient novelty. The core components of the architecture (slot-based propagation, spatiotemporal fusion, and frequency-domain enhancement) are either direct adaptations or incremental combinations of techniques that have already appeared in the literature, particularly in medical image/video segmentation and foundation model adaptation domains. Below, I detail specific concerns regarding methodology.

**Strengths:**

1. well organized and clear motivation

**Weaknesses:**

While the paper addresses an important and clinically relevant problem, the proposed method however lacks sufficient novelty. The core components of the architecture (slot-based propagation, spatiotemporal fusion, and frequency-domain enhancement) are either direct adaptations or incremental combinations of techniques that have already appeared in the literature, particularly in medical image/video segmentation and foundation model adaptation domains.

### 1. Limited novelty
The paper claims that its “context-guided slot propagation (CGSP)” mechanism is a key innovation for separating foreground and background regions in noisy echocardiographic videos. However, slot-based representations for object-centric learning and video segmentation have already been extensively studied in previous works [1–8]. The authors have not clearly articulated how this manuscript differs from or advances beyond these prior studies.

The SFF module aggregates features from reference and query frames using query-key-value attention, a pattern now very common in video segmentation such as [6,9,10-14]. For example, XMem [6] and its successors XMem++ [9] already employ cross-frame attention with memory banks to fuse spatiotemporal context efficiently. The prototype-based matching in Eq. (12) – (16) closely resembles the feature correlation and readout mechanisms in STM [11], which widely cited in video object segmentation, and the similar design can also be found in medical image domain such as [12], [13], [14]. Thus, the SFF module offers no a novel architectural or theoretical departure from established paradigms.

The FPE module applies FFT, modulates amplitude/phase with learnable masks, and uses IFFT to reconstruct features as a strategy that has seen multiple recent instantiations: [15] and [16] both exploit frequency-domain filtering or noise-robust tuning for image segmentation, explicitly addressing generalization problem in ultrasound. Frequency-aware SAM variants, such as [17], already integrate frequency priors into SAM backbones for enhanced boundary delineation under noise, which directly overlapping with the motivation of FPE.

### 2. Missing comparison with SOTA methods.

Several SOTA methods are compared under inconsistent experimental conditions:
The paper uses MiT-b2 (SegFormer backbone), which is significantly more powerful than the backbones used in many cited baselines (e.g., U-Net in early SAMUS variants, ResNet in XMem). Yet, the authors do not re-implement or re-benchmark these methods with the same backbone for a fair comparison.

### 3. Mirror Weakness
- Missing Statistical Significance and Variance Reporting
- High FLOPs and Parameter Count Undermine “Real-Time” Claim. As shown in Table 2, FESPNet has 370 GFLOPs and 34.3M parameters, which is: ~24× higher FLOPs than SimLVSeg (3G), ~3× higher FLOPs than PKEchoNet (158G) , yet only achieves marginal mDice gains. This performance even higher than Cutie (218G) and Swin-UMamba (340G), both of which are already considered heavy for real-time medical applications.

### 4. Typos
- 053 acorss frames -> across frames
- 228 a new feature map FS ∈ RK×H×W, where N is the number of slots, where is N?
- 240 ,its featurer  presentation H_{Si}∈R_L, where is L?

[1] Locatello, F., Weissenborn, D., Unterthiner, T., Mahendran, A., Heigold, G., Uszkoreit, J., Dosovitskiy, A. and Kipf, T., 2020. Object-centric learning with slot attention. Advances in neural information processing systems, 33, pp.11525-11538.

[2] Lee, M., Cho, S., Lee, D., Park, C., Lee, J. and Lee, S., 2024. Guided slot attention for unsupervised video object segmentation. In Proceedings of the IEEE/CVF conference on computer vision and pattern recognition (pp. 3807-3816).

[3] Liao, G., Jogan, M., Hussing, M., Zhang, E., Eaton, E. and Hashimoto, D.A., 2025, September. Future slot prediction for unsupervised object discovery in surgical video. In International Conference on Medical Image Computing and Computer-Assisted Intervention (pp. 219-229). Cham: Springer Nature Switzerland.

[4] Madan, S., Chaudhury, S. and Gandhi, T.K., 2024, November. Pneumonia Classification in Chest X-Ray Images Using Explainable Slot-Attention Mechanism. In International Conference on Pattern Recognition (pp. 271-286). Cham: Springer Nature Switzerland.

[5] Deng, X., Wu, H., Zeng, R. and Qin, J., 2024. Memsam: Taming segment anything model for echocardiography video segmentation. In Proceedings of the IEEE/CVF conference on computer vision and pattern recognition (pp. 9622-9631).

[6] Bekuzarov, M., Bermudez, A., Lee, J.Y. and Li, H., 2023. Xmem++: Production-level video segmentation from few annotated frames. In Proceedings of the IEEE/CVF International Conference on Computer Vision (pp. 635-644).

[7] Jaegle, A., Borgeaud, S., Alayrac, J.B., Doersch, C., Ionescu, C., Ding, D., Koppula, S., Zoran, D., Brock, A., Shelhamer, E. and Hénaff, O., 2021. Perceiver io: A general architecture for structured inputs & outputs. arXiv preprint arXiv:2107.14795.

[8] Jaegle, A., Gimeno, F., Brock, A., Vinyals, O., Zisserman, A. and Carreira, J., 2021, July. Perceiver: General perception with iterative attention. In International conference on machine learning (pp. 4651-4664). PMLR.

[9] Cheng, H.K. and Schwing, A.G., 2022, October. Xmem: Long-term video object segmentation with an atkinson-shiffrin memory model. In European conference on computer vision (pp. 640-658). Cham: Springer Nature Switzerland.

[10] Maani, F., Ukaye, A., Saadi, N., Saeed, N. and Yaqub, M., 2024. SimLVSeg: simplifying left ventricular segmentation in 2-D+ time echocardiograms with self-and weakly supervised learning. Ultrasound in Medicine & Biology, 50(12), pp.1945-1954.

[11] Oh, S.W., Lee, J.Y., Xu, N. and Kim, S.J., 2019. Video object segmentation using space-time memory networks. In Proceedings of the IEEE/CVF international conference on computer vision (pp. 9226-9235).

[12] Wang, R. and Zheng, G., 2024. PFMNet: Prototype-based feature mapping network for few-shot domain adaptation in medical image segmentation. Computerized Medical Imaging and Graphics, 116, p.102406.

[13] Yuan, Y., Wang, X., Yang, X. and Heng, P.A., 2024. Effective Semi-Supervised Medical Image Segmentation With Probabilistic Representations and Prototype Learning. IEEE Transactions on Medical Imaging.

[14] Kim, H., Hansen, S. and Kampffmeyer, M., 2025, September. Tied Prototype Model for Few-Shot Medical Image Segmentation. In International Conference on Medical Image Computing and Computer-Assisted Intervention (pp. 651-661). Cham: Springer Nature Switzerland.

[15] Chen, L., Fu, Y., Gu, L., Zheng, D. and Dai, J., 2025. Spatial frequency modulation for semantic segmentation. IEEE Transactions on Pattern Analysis and Machine Intelligence.

[16] Wei, Z., Wu, C., Du, H., Yu, R., Du, B. and Xu, Y., 2025, September. Noise-Robust Tuning of SAM for Domain Generalized Ultrasound Image Segmentation. In International Conference on Medical Image Computing and Computer-Assisted Intervention (pp. 476-486). Cham: Springer Nature Switzerland.

[17] Kim, S., Jin, P., Chen, C., Kim, K., Lyu, Z., Ren, H., Kim, S., Liu, Z., Zhong, A., Liu, T. and Li, X., 2025. MediViSTA: Medical Video Segmentation via Temporal Fusion SAM Adaptation for Echocardiography. IEEE Journal of Biomedical and Health Informatics.

**Questions:**

Please find my comments above.

---

### Meta-Review · Area_Chair_3JA7 · 2025-12-19

**Summary:**

Based on the feedback from the four reviewers, the key concerns raised are as follows: There is a limited novelty in the proposed approach, as noted by Reviewers 72j4 and 7Pap. Reviewer 72j4 pointed out the absence of comparison with strong state‑of‑the‑art baselines. Marginal performance gains over existing methods were observed by vDtk and 7Pap, raising questions about the practical significance of the contributions. The evaluation is considered insufficient: both vDtk and Z77F highlighted the lack of adequate ablation studies to isolate the contributions of individual components. Z77F further noted the performance discrepancy issue in the compared methods, suggesting possible inconsistencies in the experimental setup. Concerns about generalizability were brought up by Z77F and 7Pap, questioning whether the proposed method can maintain its performance across different datasets or application scenarios. Additionally, Z77F mentioned that some evaluation details are missing, impacting the reproducibility and clarity of the experimental section. Overall, the collective feedback indicates substantial limitations in novelty, empirical validation, and generalizability, alongside missing or incomplete comparative and ablation analyses. Given these unresolved issues, I believe the manuscript is not yet suitable for acceptance and therefore recommend rejection.

**Reviewer Concerns:**

The authors did not provide responses addressing the concerns raised by the reviewers. As a result, the key issues identified during the review process remain unaddressed.

**Reviewer Scores:**

Considering that several key concerns remain unresolved, I think the reviewers are unlikely to have a clear motivation to change their scores.

---

### Decision · Program_Chairs · 2026-01-26

Reject